# Precision Nutrition and Childhood Obesity: A Scoping Review

**DOI:** 10.3390/metabo10060235

**Published:** 2020-06-08

**Authors:** Yue Wu, Wei Perng, Karen E. Peterson

**Affiliations:** 1Department of Nutritional Sciences, University of Michigan School of Public Health, Ann Arbor, MI 48109, USA; yuewu@umich.edu (Y.W.); karenep@umich.edu (K.E.P.); 2Department of Epidemiology, University of Colorado School of Public Health, Aurora, CO 80045, USA; 3Lifecourse Epidemiology of Adiposity & Diabetes (LEAD) Center, Colorado School of Public Health, Aurora, CO 80045, USA; 4Department of Environmental Health Sciences, University of Michigan School of Public Health, Ann Arbor, MI 80045, USA

**Keywords:** life course epidemiology, DOHaD (development origins of health and disease), nutrigenomics, epigenetics/epigenomics, metabolomics, precision health

## Abstract

Environmental exposures such as nutrition during life stages with high developmental plasticity—in particular, the *in utero* period, infancy, childhood, and puberty—may have long-lasting influences on risk of chronic diseases, including obesity-related conditions that manifest as early as childhood. Yet, specific mechanisms underlying these relationships remain unclear. Here, we consider the study of ‘omics mechanisms, including nutrigenomics, epigenetics/epigenomics, and metabolomics, within a life course epidemiological framework to accomplish three objectives. First, we carried out a scoping review of population-based literature with a focus on studies that include ‘omics analyses during three sensitive periods during early life: *in utero*, infancy, and childhood. We elected to conduct a scoping review because the application of multi-‘omics and/or precision nutrition in childhood obesity prevention and treatment is relatively recent, and identifying knowledge gaps can expedite future research. Second, concomitant with the literature review, we discuss the relevance and plausibility of biological mechanisms that may underlie early origins of childhood obesity identified by studies to date. Finally, we identify current research limitations and future opportunities for application of multi-‘omics in precision nutrition/health practice.

## 1. Introduction

### 1.1. Childhood Obesity: A Persistent Public Health Challenge

Child obesity remains a persistent public health concern worldwide, with little evidence that countries or regions with highest rates have been able to reverse trends [1]. Recent data from Centers for Disease Control and Prevention (CDC) in the United States documented a prevalence of childhood obesity of 18.5% in 2015–2016, indicating nearly one in five children were obese [2,3,4,5]. These burgeoning trends have spurred widespread public health efforts to reduce and manage the childhood obesity pandemic [6,7], framed by the social-ecological approaches [8] that emphasize multiple levels of influence, ranging from individual and family to organizational and community/policy approaches, as well as social determinants of health [8]. Complementary frameworks integrate a developmental perspective, known as the Six-Cs model. This model highlights that not only environmental but also hereditary factors have potential to influence disease risk at any stage of child development from infancy through adolescence [9,10,11]. Such frameworks acknowledge mediating and moderating effects among different levels of risk factors [10] and align with central concepts of developmental origins of health and disease and life course epidemiology.

### 1.2. Developmental Origins of Obesity-Related Disease: A Focus on Early Life as a Window of Opportunity for Modifications Based on Life Course Frameworks

Over four decades ago, Ravelli and colleagues tested the hypothesis that prenatal and early postnatal nutrition are determinants of subsequent obesity, using historical data from 300,000 young adult men exposed to the Dutch famine of 1944–1945 during the *in utero* period [12,13,14]. Perhaps the most noteworthy finding was that the direction and magnitude of associations between *in utero* famine exposure and future obesity risk depended on timing of exposure. These initial findings, in conjunction with those of others that ensued—including two population-based studies led by David Barker showing an inverse relationship between weight in infancy and adult ischemic heart disease mortality [15,16]—put forth the “Fetal Origins of Adult Disease (FOAD)” hypothesis that was subsequently formalized as “Development Origins of Health and Disease (DOHaD)”.

While FOAD focused on the pre-, peri-, and immediate postnatal periods (typically spanning birth through the first 2 years of life) as key life stages of interest for “programming” future health, DOHaD recognized the relevance of critical/sensitive periods beyond infancy including: early childhood around the time of the adiposity rebound [17]; late childhood/early adolescence when the number of adipocytes an individual possesses is thought to be established for life [18]; and the pubertal transition [19], a time of rapid growth and hormonal fluctuation. Pregnancy is now hypothesized as another sensitive period not only for the fetus but also for the gravida given findings that adverse pregnancy outcomes (e.g., gestational diabetes, hypertensive disorders, preterm birth) presage a woman’s as well as her offspring’s risk for cardiometabolic disease down the road [20,21]. However, to our knowledge, there is no existing population-based epidemiological multi-‘omics studies that focus on childhood obesity as the health outcome, which could expand our knowledge on this field.

In parallel with growing interest in DOHaD, life course epidemiology rose to prominence as a methodological causal inference framework to study DOHaD [13,22,23,24]. Additionally, advancements in high-throughput technologies enabled researchers to study holistic biological mechanisms that may underlie DOHaD findings. While the focus has been on epigenetics and epigenomics, which involve the study of the complete set of potentially reversible epigenetic modifications made to the DNA sequence [25], other ‘omics fields are now receiving attention: nutrigenomics (the study of gene and nutrient interactions) [26] and metabolomics (the study of low-molecular-weight compounds in biological tissues) [27].

In this scoping review, we consider the literature surrounding ‘omics mechanisms identified during three sensitive periods that may influence the development of childhood obesity: (1) *in utero*, including germinal, embryonic, and fetal periods; (2) infancy, ranging from birth to 2 years of age; (3) early-childhood, ranging from 2 to 6 years of age. The endpoint of interest is obesity during middle childhood, generally considered as ages 7–12 years. Prevalence of obesity increases markedly in mid-childhood [28,29,30,31], coincident with peak weight velocity and changes in adipose tissue distribution [32] that drive pubertal onset. The objectives are threefold. First, we summarized current literature surrounding evidence from three ‘omics fields (nutrigenomics, epigenetics/epigenomics, and metabolomics) that link early life nutritional exposures to obesity in mid-childhood. Secondly, while many factors are involved in the development of excess adiposity, we describe specific mechanisms identified in the literature as potential pathways linking nutrition to overweight/obesity during childhood. Finally, we point out weaknesses and chasms in current literature, and make recommendations for future directions in the realm of precision nutrition/health field—a relatively new initiative that seeks to elucidate inter-individual variability in disease risk factors and mechanisms.

## 2. Methodology

We elected to conduct a scoping review because the application of precision nutrition in childhood obesity prevention and treatment is relatively recent, underscoring the importance of identifying knowledge gaps to expedite further studies in this field [33]. Evidence from original research included in this scoping review was identified using Google Scholar, Medline, EMBASE, and DynaMed databases. First, we conducted a comprehensive search effort to gather materials relevant to the main objective of this review—the potential associations between nutrition, ‘omics, and childhood obesity. We then considered population-based, peer-reviewed studies that were conducted and published within the last 10 years, including observational studies with a range of designs (cross-sectional, case control, and prospective longitudinal studies). We did not limit dates or types (in vivo/in vitro/human studies) for literature we cited to interpret the associations relating ‘omics evidence of *in utero*, infancy, and early childhood exposure to risk of obesity in mid-childhood obesity. Key search terms included “DOHaD theory,” “DOHaD,” “thrifty theory,” “maternal/*in utero* nutrition exposure,” “BMI (Body mass index) rebound,” “infancy, breastfeeding,” “infancy, nutrition intake,” “early childhood,” “early childhood, nutrition intake,” “early childhood, dietary behavior,” “precision nutrition/medicine,” “omics networks,” “multi-‘omics networks,” “childhood overweight/obesity” in combination with any of “nutrigenomics,” “nutrition and genetic interactions,” “epigenetics,” “DNA methylation,” “methylome,” “histone modification,” “metabolites,” or “metabolome”. To comprehensively identify both current limitations and future research opportunities, we expanded the literature search to include papers considering relevant methods, including cohort selection, data collection, statistical analysis, underlying multi-‘omics networks. Our search identified 67 peer-reviewed articles (not including methodologic literature). Among these, 23 articles were reviewed and discussed in detail and contributed to our focus on linkages between nutrition exposure —> ‘omics —> childhood obesity pathways (Table 1). Summary and discussion of reviewed articles were organized and presented by types of ‘omics, e.g., nutrigenomics, epigenomics, and metabolomics. Within each ‘omics theme, results were aggregated by sensitive periods (*in utero*, infancy, and early-childhood).

## 3. ‘Omics Fields of Interest in This Review

The oldest and most well-known ‘omics is genomics, or the study of DNA content within a cell to understand the structure and function of an organism’s genome. In this review, we re-route the focus to nutrigenomics, also known as the study that examines gene-diet interactions—a rapidly-growing field that seeks to study the interaction between genetics (including genome-wide measures) and nutrition to identify the optimal diet for a person’s genetic profile [26]. Following genomics is epigenomics, or the study of the complete set of epigenetic modifications on top of an organism’s DNA sequence. While it is debated whether epigenomics should be formally considered as an ‘omics field given that there is no molecular product, we elected to review the epigenomics literature in relation to childhood obesity given the relevance of epigenetic mechanisms in the maintenance of phenotypic plasticity [56]. Following epigenomics are transcriptomics, the study of RNA transcript; and proteomics, study of the structure and function of proteins. While transcription is functionally relevant to gene expression, and proteomics facilitate understanding of how proteins carry out basic life functions, we do not review these two ‘omics given the sparse literature. Finally, at the end of the ‘omics cascade and closest to our phenotype is metabolomics, the systematic study of low-molecular-weight compounds in biological tissues and fluids. As metabolites are a synthesis of processes that occur both inside and outside of the body, metabolomics has potential to reveal novel insights into mechanisms and pathways of childhood obesity [57]. Of note, despite increasing evidence of a role for the gut microbiome as a modifier or mediator to ‘omics mechanisms, we did not include this topic in the present review since the microbiome is not typically considered part of the ‘omics cascade, and because composition of the gut microbiota can be affected by weight status and thus, may be considered anoutcome rather than a mechanism.

## 4. Summary and Discussion of Findings

### 4.1. Nutrigenomics

Nutrigenomics is the holistic study of gene–diet interactions. This field followed that of the older concept of nutrigenetics, which has been sensationalized in popular press as “feeding your DNA”, or “eating for your genes.” Nutrigenetics scrutinizes the effect of polymorphisms of single genes and how these differences affect one’s metabolism and thus, carries implications for the most appropriate foods to consume (or avoid). On the other hand, nutrigenomics is concerned with the potential for diet to change gene function and expression via alterations of the epigenome. Indeed, the goal of nutrigenomics is to unravel the interaction between genetics and dietary intake by bringing together bioinformatics, nutrition, molecular biology, genetics, genomics, epidemiology, and molecular medicine [26]. In this review, we focus on nutrigenomics and note that because most studies to date have only assessed diet and the epigenome at a single point in time, it has not been possible to untangle whether changes in diet cause changes in the epigenome. Thus, current literature simply provides evidence for an association between dietary intake and genome-wide differences in DNA methylation.

From a nutrigenomics perspective, nutrients are dietary signals that are detected by the cellular sensory systems that influence gene and protein expression and, subsequently, metabolic consequences [58]. The effect of gene-diet interaction on risk of childhood obesity has been observed in both animal- and human-based studies [59]. In the studies that focus on the association between maternal nutritional programming of fetal adipose tissue development and long-term consequences of later obesity, a U-shaped relationship has been observed between birth weight and BMI later in life. The connection between birth weight and adult weight suggests that there are enduring effects of the *in utero* environment on later obesity risk [60]. Conversely, maternal nutrient deprivation during late fetal development could result in reduction of offspring birth weight, as well as glucose intolerance and insulin resistance later in life, which also are risk factors of childhood obesity [61]. This finding is in accordance with the “thrifty genotype hypothesis”, which implies that gene variants may lead to an increased energy deposition as fat accumulates over time to maintain reproductive function and enhance survival [62]. Research activities in the field of nutrigenomics have two major foci: (1) identify the genes that respond to diet and nutrition modifications; and (2) examine the interaction between diet and nutrition modifications with metabolic homeostasis [58]. We summarize the literature according to these two categories, below.

#### 4.1.1. Nutrigenomics during the *In Utero* Period and Childhood Obesity

Most nutrigenomics-obesity research during the *in utero* period comprises genome-wide association studies (GWAS) and candidate gene analysis recognized for their potential roles in maternal carbohydrate, lipid metabolism, insulin resistance, or energy expenditure regulation [63]. However, whether the “hits” identified in these analyses are related to offspring health outcomes, including childhood obesity, is often left unexplored.

In Joffe and Houghton’s review [64], they specifically subdivided obesity into monogenic and polygenic obesity types. Monogenic obesity refers to rare forms of severe obesity that result from polymorphisms that carry a large effect size on an individual gene or chromosomal region. Polygenic obesity proposes that frequent alleles of several genes, each with a small effect on the BMI (or weight-related phenotype), determine an individual’s weight by the sum of the variants present [64]. The complexity of childhood obesity itself, with the challenges of collecting and analyzing food intakes, make population-based nutrigenomics difficult, yet valuable.

Kaput [65] presented a comprehensive list of genes analyzed in response to nutrition or in nutrition associations, but none of these studies included obesity as their research outcome. Thisconsitutes a common limitation of population-based nutrigenomics studies—in addition to identifying gene and gene-pathways of interest, there is a need to link these pathways to nutritional factors and childhood obesity. For example, Vincent et al. [34] showed the gene-diet interactions between genes *Apoprotein E, Apoprotein B,* and intestinal fatty acid-binding protein (*FABP*) with lipoprotein metabolism using the RIVAGE project cohort, which evaluates the health effects of a Mediterranean-type (MED) diet and a low-fat low cholesterol (LFLC) diet. Since the association between lipid metabolism and childhood obesity is well-demonstrated [35], we presumably can draw an association between types of fat intake, expression of genes *Apoprotein E, Apoprotein B*, and risk for childhood obesity. For animal-based studies, this type of study is easier to design and conduct, as the short life span of mouse models, availability of genetically designed knockout mouse and dietary variables than can be more easily manipulated with the experimental design [66,67].

In more recent nutrigenomics studies, researchers have identified inter-individual, genotype-dependent differences in the response to dietary intake, which translated to variability in individual response to the same foods [68]. For example, Cook and Fletcher [36] used sibling pairs with genetic data from the Wisconsin Longitudinal Study cohort to explore the interaction between the fatty acid desaturase-2 (*FADS2*) gene and essential fatty acids metabolism, as well as cognitive development outcome. Their results did not show a significant direct interaction between the *FADS2* gene and early life nutrition that eventually would affect later-life IQ. However, they did observe a positive and statistically significant interaction between *FADS2* and birth weight [36]. Considering *FADS2* gene is involved in regulation of unsaturation of fatty acids [37], we can hypothesize that *FADS2* may also affect the metabolism of dietary fatty acid intake, and ultimately, childhood weight status.

#### 4.1.2. Nutrigenomics during Infancy and Childhood Obesity

The author Adolph P. Gouthey is famous for this quote, “*If life were measured by accomplishments, most of us would die in infancy.*” Placed in the context of DOHaD, infancy is indeed the most developmentally plastic stage characterized by rapid growth and development [69]. Thus, it should come as no surprise that many ‘omics studies have focused on the infancy period as a key life stage of interest to understand mechanisms linking nutritional exposures to obesity-related health outcomes.

With respect to nutrigenomics, relevant studies have focused on genes involved in the central control of energy balance, adipogenesis, lipid turnover, adaptive thermogenesis, oxidative metabolism, insulin signaling, energy metabolism, and lactation behaviors [70,71]. For example, genes related to leptin (*LEP*) expression and regulation have been of particular interest given the physiological role of breast milk leptin in body weight control among developing infants. Leptin is a hormone primarily produced and secreted by the adipose tissue, and its circulating levels signals nutritional status and energy storage levels to feeding centers in the hypothalamus and other central areas [72,73]. Biological metabolism pathways have shown that leptin signals the expression and release of orexigenic neuropeptide Y (*NPY*), and pro-opiomelanocortin (*POMC*), amino acids that are involved in various physiological and homeostatic processes in both the central and peripheral nervous systems [74]. *NPY* and *POMC* induce the expression of leptin receptor (OB-Rb), which activates the transcription intracellular signaling cascade and produces an endogenous inhibitor to suppress the expression of cytokine signally 3 (*SOCS3*) gene [75] Overexpressed *SOCS3* contributes to both leptin resistance and insulin resistance; thus, suppressed *SOCS3* gene expression prevents against insulin resistance and obesity [76]. These biological findings are in accordance with leptin-involved breastfeeding behavior and childhood obesity outcomes in human studies. Miralles et al. [38] recruited a group of 28 non-obese women who breast-fed their infants for at least 6 months. During the whole lactation period, venous blood and milk samples were obtained from mothers at 1, 3, 6, and 9 months; and infant body weight and height were followed until 2 years of age. They found that milk leptin concentration at 1 month of lactation was negatively correlated with infants’ BMI at 18 and 24 months of age, and log-transformed milk leptin concentration at 1 and at 3 months was negatively correlated with infants’ BMI from 12 to 24 months of age [38]. Therefore, presumably, when the *NPY* and *POMC* genes are active, appropriate regulation of food intake and energy expenditure enable infants to maintain the amount of fat stores within a certain healthy range in a longer term [39,77].

Several studies have also revealed that genetic variation in apolipoprotein (apo) genes (*apoA-I, apoA-IV, apoB, apoCIII, apoE*), and LDL-receptor genes were involved in modulating the response to diets with different calorie or lipid concentrations [40,41]. Prospective twin studies have also supported the interaction of genetic variability with dietary responsiveness, as well as its essential role in affecting childhood obesity. Poehlman et al. [78] conducted the first human study that demonstrated the individual variation in the response to overfeeding in total calories, as well as the sensitivity of adaptive body fat changes under the influence of genetics. They recruited six pairs of male monozygotic twins matched on age to investigate the effects of a 22-day overfeeding on the body composition and fat morphology characteristics. At the end of the study period, they observed significant changes with strong within-pair resemblance in body weight, fat mass, sum of skin folds, and fat cell diameter [78]. Hainer et al. [42] not only examined the intra-pair correlation, but also inter-pair discrepancies. In this analysis, the researchers recruited 14 pairs of female monozygotic twins to complete a 4-week very-low-calorie diet, and observed change of body weight and body fat. The authors reported greater variability (12–17-fold) in body weight and fat mass fluctuation between vs. within pairs, suggesting that gene–diet interactions are drivers of physiological responses to fat and cholesterol intake, as well as total caloric intake. Thus, it is not surprising that short-lived, popular diets often engender very different responses in different individuals [42]. In the future, population-based nutrigenomics research that not only identifies genes responsible for inter-individual variability in physiological response to various diets, but also investigates how such genes may modify the association between diet and obesity-related outcomes during early life are warranted.

#### 4.1.3. Nutrigenomics during Early Childhood and Childhood Obesity

When reviewing the literature, we identified very few population-based cohort studies examining the potential association of nutrigenomics in relation to obesity during early childhood. Such studies have been summarized in a few reviews that list the genes associated with nutrition related obesity phenotypes in children [79]. Key findings from these reviews include consistent associations of lower *LEP*, *LEPR*, *POMC*, *MC4R* gene expression in relation to higher food intakes [66]; variability in polymorphisms of the *ACP1, ADRB3*, and *UCP2* genes in relation to energy metabolism efficiency; and disrupted *PPARG*, *INSIG2,* and *INS* gene expression with respect to glucose-insulin homeostasis, and lipid metabolism [79].

To date, the only human nutrigenomics study we found that explores the entire pathway of diet → genetic variant→ obesity-related health outcomes in children was led by Qi et al. [43]. In this study, the researchers observed a biological interaction between the *FTO* variant and dietary protein intake on BMI. Specifically, the association between *FTO* genotype and BMI was much stronger in children with high protein intake than in those with low intake. Thus, their results suggested the *FTO* variant that confers a predisposition to higher BMI is associated with higher total energy intake, while lower dietary protein intake attenuated the association between *FTO* genotype and adiposity [43]. However, this study was a combined analysis of 16,094 boys and girls aged 1–18 years from 14 studies, which may not be appropriate given marked variability in physiology across this age range.

### 4.2. Epigenetics

Epigenetics, or the study of mitotically heritable yet reversible molecular modifications to DNA and chromatin without alterations to the underlying DNA sequence [25], is recognized as an important biological pathway underlying DOHaD phenomena [80]. Epigenetic modifications, including DNA methylation, histone acetylation, and the activities of small non-coding RNAs, are basic physiological processes that underlie mammalian multicellular organismal development, from conception to cellular/tissue differentiation to aging and across generations via the germ-line cells [81].

A series of analyses that transpired from the Dutch Famine Birth Cohort Study provided strong evidence that adverse nutritional conditions (namely, famine) early in development is a risk factor for cardiometabolic disease later in life. Researchers have found that exposure to famine at any stage of fetal development tends to have reduced glucose tolerance, raised insulin concentration, higher blood pressure, higher BMI, and more atherogenic plasma lipid profile [82]. Heijmanss et al. [83] further examined the mechanisms behind these relationships and found that individuals who were prenatally exposed to famine during the Dutch Hunger Winter had less DNA methylation of the imprinted insulin-like growth factor II (*IGF2*) compared with their unexposed, same-sex siblings six decades later.

Since these initial studies that employed candidate gene approaches, technology to investigate epigenetic changes has markedly improved afterwards. Currently, the three major approaches to assessment of DNA methylation used in epidemiological research are global DNA methylation, gene-specific DNA methylation, and epigenome-wide studies of DNA methylation at the resolution of single CpG sites. In studies of pediatric populations, DNA methylation has largely been assayed from circulating leukocytes given challenges of more invasive sampling procedures to obtain target tissues (e.g., adipose tissue) [84]. In studies published through the early 2010s, a large emphasis was placed on associations of early nutritional exposures and obesity-related outcomes in relation to global DNA methylation—an indicator of total methyl group content within an individual’s genome, and a metric of genomic stability [85,86,87]. However, a limitation to both global DNA methylation and candidate gene studies is that the etiology of complex diseases (such as obesity and related conditions) likely involves several biological pathways. This notion gave rise to use of epigenome-wide association studies (EWAS) to identify multiple differentially methylated regions of the genome to gain a more holistic understanding of epigenetic alterations associated with various exposures or health outcomes [88,89]. In the subsequent sections, we discuss existing literature as it pertains to each of the three methods of assessing DNA methylation (global DNA methylation, candidate gene DNA methylation, and EWAS).

#### 4.2.1. Epigenetics during the *In Utero* Period and Childhood Obesity

The relevance of epigenetics in DOHaD stems from the fact that sensitive windows of development coincide with higher lability of the epigenome. For instance, shortly after fertilization, a genome-wide demethylation event occurs, followed by systematic re-establishment of DNA methylation [90]. Accordingly, the *in utero* period is a well-studied sensitive developmental phase in the field of DOHaD [91].

One of the major limitations of ‘omics research is that very few human studies have employed the appropriate study design or sufficient length of follow-up to examine the dietary exposure, ‘omics, and health outcome in the same population. In a review conducted by Reynolds et al. [92], the authors thoroughly summarized current evidence on the association of maternal exposures with DNA methylation of certain genes, and the association of DNA methylation with health outcomes. Genes of interested included but were not limited to: *IGF2*, *ADIPOQ*, *NR3C1, SLC6A4, MEG3, CYP1A1.* However, these genes were not then linked to modifiable exposures, like diet, or to important offspring health outcomes.

In addition to Reynolds et al.’s review, several original research papers have been published. Soubry et al. [93,94] examined parental preconceptional obesity, which is a known obesogenic, diet-related exposure during the *in utero* period [44], in relation to DNA methylation profiles among 92 newborns enrolled in the prospective Newborn Epigenetics STudy (NEST) cohort. After adjusting for potential confounders and cluster effects, paternal obesity was significantly associated with lower methylation levels at multiple human imprinted genes that are important in growth and development, including DMRs of maternally expressed gene 3 (*MEG3*), necdin (*NDN*), small nuclear ribonucleoprotein polypeptide N (*SNRPN*) and epsilon sarcoglycan/paternally expressed gene 10 (*SGCE/PEG10*). Another study conducted by Liu et al. [45] using the same cohort, among 320 women with blood drawn at gestational age < 120 d, found that higher birth weight of their offspring was associated with higher DNA methylation levels at the *SGCE/PEG10* DMRs, and lower methylation levels at the *MEG3* DMRs. Connecting these two pieces of evidence together, we may be able to deduce that epigenetics mechanism, specifically DNA methylation of *SGCE/PEG10* and *MEG3* DMRs, provide an interface between paternal obesity and higher birth weight, a risk factor for later childhood obesity.

With the development of DNA extraction, immunoprecipitation, and sequencing technologies, scientists have been able to examine the epigenetic effects at a finer resolution. Three notable population studies have evaluated the direct effect of prenatal epigenetic modifications on childhood obesity presence at genome-wide levels. First, DNA methylation at two Insulin-like Growth Factor 2 (*IGF2*)/*H19* DMRs collected from umbilical cord blood leukocytes of 204 newborns was higher in children with weight-for-age (WFA) > 85th percentile than in those with WFA ≤ 85th percentile [95]. Similarly, Relton et al. observed that DNA methylation at caspase 10 (*CASP10_P2*), cyclin-dependent kinase inhibitor 1C (*CDKN1C_P2*), ephrin type-A receptor 1 (*EPHA1_P*), HLA class II histocompatibility antigen DO beta chain (*HLA_DOB3*), matrix metalloproteinase 9 (*MMP9_P*), myeloproliferative leukemia virus oncogene (*MPL_P*), and nidogen-2 (*NID1_P*) quantified from 24 cord blood samples, were associated with at least one weight-related index (BMI, fat mass, lean mass) later in childhood [96]. Lastly, Godfrey et al. measured the methylation status of 68 GpGs 5′ from five candidate genes in umbilical cord tissue DNA from 455 neonates; and found higher methylation of retinoid X receptor-α (*RXRA*) chr9 was associated with higher total fat mass and fat percentage among those subjects in childhood [97]. However, as mentioned before, these examples did not assess *in utero* exposures that may lead to differences in DNA methylation.

#### 4.2.2. Epigenetics during Infancy and Childhood Obesity

Among infants, a key area of interest considers epigenetics as a consequence of mode of feeding (i.e., breastfeeding duration and exclusivity, formula vs. breast milk, and breast- vs. or bottle-feeding) [98], with respect to birth size. Another area of influence highlights the relationships between modifications of epigenetic and cardiometabolic health outcomes.

Feeding mode and birth size. Regarding feeding mode, leptin is an essential hormone that can never be overlooked. As stated above, animal and human studies have shown that leptin from breast milk plays a physiological role in body weight control among developing infants [72,73]. The underlying rationale is that the functionality of leptin, a 16 kD peptide hormone encoded by the gene *LEP* is involved in adipose mass lipogenesis and appetite regulation [46,47,99], both of which have direct implications on weight control and obesity risk. For instance, Lesseur et al. [48] collected blood samples at different time points from 39 mother–infant pairs. They observed a strong correlation between maternal blood *LEP* promoter DNA methylation and infant blood *LEP* methylation; and secondly, noted higher *LEP* promoter DNA methylations among small-for-gestational-age (SGA) infants only. Higher *LEP* promotor DNA methylation could result in lower circulating leptin levels, then affect infant growth and development. These effects seem independent of insulin and the growth hormone insulin-like growth factor-I system [100], and have important implications given that SGA has been linked to elevated risk of obesity during childhood.

Cardiometabolic profile. Many epigenetic studies in childhood obesity research have focused on genes that could be linked with insulin activity and lipid metabolism, as these are established risk factors of overweight and obesity. Rzehak et al. [101] conducted a EWAS based on a subset of 374 children out of 543 children aged 5.5 years from European Childhood Obesity Trial Study (CHOP) cohort, examining the potential association between epigenetic programming and childhood body composition and obesity. After Bonferroni correction, they have found 13 genes in which DNA methylation variants were significantly associated with outcomes of interest, and they were *SNED1* (Sushi, nidogen and EGF like domains), *KLHL6* (Kelch like family member 6), *WDR51A* (WD repeat domain 51A), *CYTH4-ELFN2* (cytohesin 4, extracellular leucine-rich repeat and fibronectin type III domain containing 2), *CFLAR* (CASP8 and FADD like apoptosis regulator), *PRDM14* (PR/SET domain 14, previously PR domain containing 4), *SOS1* (SOS Ras/Rac guanine nucleotide exchange factor 1), *ZNF643* (Zinc finger protein 643), *ST6GAL1* (ST6 beta-galactosamine alpha-2, 6-sialyltransferase 1), *C3orf70* (Chromosome 3 open reading frame 70), *LOC101929268* (uncharacterized noncoding RNA), *MLLT4* (myeloid/lymphoid or mixed-lineage leukemia) and *CILP* (Cartilage intermediate layer protein 2). These genes are either closely related to insulin sensitivity or brown and white adipose tissue formation [101], which supports the hypothesis that the interactions between dietary exposure and insulin- and lipid-metabolism involved epigenetic changes have possible impact on childhood obesity.

#### 4.2.3. Epigenetics during Early Childhood and Childhood Obesity

To our knowledge, all published population-based epigenetics studies that focused on early childhood period have been either cross-sectional, or no study to date has gone the additional step of linking findings to dietary or nutritional exposures. As an example, Fradin et al. [102] conducted a genome-wide methylation case-control analysis searching for methylation marks associated with obesity in children, aged ranged from 3 to 13 years old. They identified, respectively, 18 and 138 at 10 gene loci differentially methylated CpG sites between moderate or severe obese children and lean controls, respectively, after correction for cellular heterogeneity and multiple testing [102]. They noted that the majority of these genes in variations are involved with growth factor signaling, inflammation status, or obesity-associated hormone production [102]. Similarly, Ding et al. performed a comprehensive genome-wide screen of DNA methylation, trying to identify novel markers in childhood obesity [103]. Thirty-two obese early childhood children aged from 3 to 6 years and equal numbers of age- and gender-matched lean children were included in the study. They observed that compared to lean children, 251 promoter CpG islands were demethylated and 141 were hypermethylated in obese children [103]. In addition, Wang et al. observed significantly higher hypoxia inducible factor 3 alpha subunit (*HIF3A*) methylation levels in 110 obese children compared to 110 normal-weight, age-, and gender-matched controls, demonstrating the positive association of *HIF3A* DNA methylation with BMI and BMI change [104].

### 4.3. Metabolomics

Metabolomics is the large-scale study of low-molecular-weight compounds in biological tissues and fluids and thus, considered the closest to the phenotype of obesity. In large epidemiological studies, the most common bio-tissue of interest is blood given that it is relatively easy to obtain and minimally invasive. While the measurement of metabolomics in blood is analogous to assessment of metabolic biomarkers indicative of obesity-related disease risk, such as insulin, glucose, inflammatory cytokines, the novelty is the ability of high-throughput platforms via nuclear magnetic resonance (NMR) spectroscopy or mass spectrometry (MS) to systematically and comprehensively quantify thousands of metabolites at a time [27,105]. As these compounds in blood are both responsive to environmental cues as well as internal physiology, circulating metabolites are thought to integrate processes occurring both inside as well as outside of the body. Accordingly, metabolites have potential to provide information on disease processes and risk, as well as environmental exposures to target in order to prevent disease.

Xie et al. [49] published a literature review of animal-based studies that sought to identify potential mechanisms of metabolites involved in obesity or obesity-related sequelae. Pathways of interest included (1) glucose metabolism and tricarboxylic acid (TCA) cycle, (2) lipid metabolism, (3) choline metabolism, (4) amino acid metabolism, and (5) creatine metabolism. The study of metabolomics in humans is inherently more challenging due to intra- and inter-individual variability in metabolism, as well as difficulties in collection of ideal biospecimens. In the following sections, we will continue to discuss population-based studies that examined and evaluated the impact of the change of metabolites on childhood obesity risks, through modifying nutritional factors while pointing out limitations and gaps in specific literature.

#### 4.3.1. Metabolomics during the *In Utero* Period and Childhood Obesity

Metabolomics profiling of maternal serum during pregnancy has been used to shed light on the *in utero* environment. Klebanoff and Keim [51] studied 1986 mother-child pairs to examine the association between caffeine metabolites, paraxanthine collected at <20 and ≥26 weeks’ gestation in maternal serum, and relative risks of obesity (BMI ≥ 95th percentile for age and sex) measured among children who were followed with visits at 48 and/or 84 months of age. They observed a modest, negative linear association between paraxanthine at ≥26 weeks with obesity at age 7 years, after adjusting for a list of maternal confounders [51]. Such findings suggest a potential link between maternal metabolism of exogenous chemicals like caffeine and future risk of obesity in offspring several years later. However, we note that while maternal blood may provide a measure of the intrauterine metabolic milieu, a limitation of this particular biospecimen is that it is not possible to determine the extent to which circulating metabolites in maternal blood reflect fetal exposures given that the placental is an important interface between maternal condition and the *in utero* environment. Thus, inference on fetal programming via metabolomics of maternal blood is limited.

In addition to metabolites that affect energy production, compounds involved in nucleic turnover have also been of interest. Rauschert et al. [50] also presented the evidence showing the associations between fatty acid metabolites and risks of childhood obesity. Multiple studies have confirmed alterations in serum long chain polyunsaturated fatty acid (LC-PUFA) as a link between maternal BMI and offspring obesity. Specifically, Donahue et al. [106] showed a negative association between maternal omega-3 fatty acids metabolites and offspring adiposity. However, a key challenge when exploring pathways that may, for example, link maternal weight status to offspring weight status is ascertaining the extent to which these metabolites reflect true disturbances in metabolism due to environmental factors that affect both maternal condition and offspring condition vs. whether they simply reflect shared genes. Clever study designs to minimize the impact of genetics include comparing effect sizes of associations for maternal vs. paternal weight status [107], conduct within- vs. between-family comparisons [108], and leverage sibship analyses [109].

#### 4.3.2. Metabolomics during Infancy and Childhood Obesity

Given the difficulty of collecting biospecimens from newborns, umbilical cord blood is used as a common surrogate for the infant metabolome. Perng et al., [110] identified the associations of cord blood metabolite patterns with birth size, which is an early indicator of childhood obesity [111]. These authors quantified metabolites in cord blood of 126 mother–child pairs, and collected gestational age- and sex-standardized birth weight-for-gestational age z-scores. They found that comprised metabolites involved in energy production (malate, succinate, fumarate) and nucleic turnover (inosine 5-monophosphate, adenosine 5-monophosphate, cytidine 5-monophosphate) were associated with higher birth weight-for-gestational age z-scores at birth. Similarly, Isganaitis et al. [112] performed a nested case–control study to explore the associations of cord blood metabolites with early childhood obesity risk, using a longitudinal cohort of mothers and children pairs. They found several cord blood metabolites were associated with rapid postnatal weight gain, including three metabolites that are related to tryptophan (serotonin, tryptophan betaine, and tryptophyl leucine), two methyl donors (dimethylglycine and *N*-acetylmethionine) and glutamine to glutamate ratio. However, a key weakness of these studies is that they assessed cord blood metabolites at a single point in time, which precludes inference on upregulation vs. downregulation of specific pathways, or differences in metabolite concentrations across the maternal/fetal unit.

Some investigators have been able to collect infant blood. Kirchberg et al. [52] collected plasma samples of 691 infants who received formula milk with different protein content (high protein vs. low protein), then their changes in amino acid and acylcarnitine concentrations were quantified. There were 29 of the 62 analyzed metabolites that differed significantly between the high protein and low protein formula groups. Specifically, branched chain amino acids (BCAAs) and their degradation products, the short-chain acylcarnitines C3, C4, and C5 were significantly elevated in the high protein formula group. In comparison, long-chain acylcarnitines were decreased among high protein formula fed infants [52]. Since a high protein intake during infancy could result in elevated insulin and *IGF-1* levels, high protein formula fed infants are at higher risk of obesity in childhood [53].

#### 4.3.3. Metabolomics during Early Childhood and Childhood Obesity

To date, most metabolomics studies in children followed in the footsteps of findings from adult populations, and focused on targeted analyses of metabolites involved in amino acid and fatty acid metabolism given findings in adults [54,55,113]. In one of the first metabolomics studies in children, Michaliszyn et al. [114] found that among 139 obese and normal weight children, there was a positive association of BCAA metabolites with beta-cell function relative to insulin resistance. Moran-Ramos et al. [115] and McCormack et al. [116] also observed elevated BCAAs and aromatic amino acids (AAAs) metabolites in obese children compared with normal weight children. In a 2013 study, Perng et al. leveraged untargeted metabolomics profiling, which provides a snapshot of all detectable metabolites in a biospecimen, in a cohort of youth in Boston to identify metabolomics profiles associated with obesity and metabolic risk at ages 10 years [117]. The metabolomics profiles detected included the BCAA metabolite pattern, as well as an androgen hormone metabolite pattern involved in steroid hormone synthesis, both of which were associated with intake of fast-foods and obesity status and several metabolic risk biomarkers [117]. The researchers subsequently followed up these children through early adolescence and found that both patterns at age 6–10 years were associated with adverse changes in conventional metabolic biomarkers during 5 years of follow-up [117], providing some evidence that certain metabolite profiles may precede worsening of metabolic health in youth. Building on on these findings, Perng, et. al., explored metabolomics profiles of childhood obesity and metabolic risk in diverse populations, including in a cohort of Mexican adolescents (with a median age lying in the mid-childhood range: 7.7 years (range: 6.7–10.6)). Authors identified metabolites involved in lipid, amino acid, and carbohydrate metabolism as correlates of a metabolic syndrome risk score [118]; confirmed that the BCAA metabolite pattern is associated with prospective changes in glycemia and lipid biomarkers during the adolescent transition [119]; and identified urate and nonanoate as potential markers linking intake of sugar sweetened beverages to higher blood pressure [120]. Together, these studies point to the utility of metabolomics to not only shed light on mechanisms underlying obesity and metabolics risk, but also show feasibility of linking metabolites to aspects of diet in children.

### 4.4. Challenges Surrounding Collection of Biospecimens and Dietary Data

Despite consensus that more ‘omics studies in infants, children, and adolescents are needed, a major obstacle of nutrigenomics, epigenetics, and metabolomics analyses revolve around challenges of obtaining the most appropriate biospecimens. For example, when conducting metabolomics studies during pregnancy, it may not only be difficult to obtain fasting blood from women in order to characterize the *in utero* environment, but also, it is difficult to know whether the maternal metabolome is an accurate depiction of the metabolic milieu experienced by the fetus given the checks and balances put in place by the placenta. Similarly, while blood from infants would be an ideal way to assess an infant’s epigenome and metabolome, drawing blood from young infants is quite challenging and researchers may rely instead on cord blood which presents its own set of unique challenges of interpretation. In young children, blood draw can cause significant emotional burden to the study subjects and their parents. Thus, additional research is needed to understand whether and to what extent ‘omics analyses of less invasive biospecimens (e.g., urine, saliva, hair, nails, skin cells) correlate with what is currently known with respect to ‘omics analyses of blood. Secondly, despite the fact that infancy and early childhood are key life stages that shape life-long dietary habits and preferences [121,122,123], gathering long-term dietary information from infants and young children—even with assistance from caregivers—is an ongoing challenge. Thus, of the goal of achieving better understanding of the relationships among early life nutrition, ‘omics, and obesity-related health outcomes is truly an interdisciplinary effort that requires communication and collaboration across multiple fields.

## 5. Research Gaps and Future Directions

### 5.1. Summary of Research Gaps

In this review, we described studies that sought to identify biological mechanisms linking early nutrition to obesity during the *in utero* period, infancy, and childhood. We identified three major gaps in current literature.

First, early childhood (ages 2–5 years) is an essential development period, yet very few studies have focused on this timeframe. Potential reasons include: (1) perceptions that childhood is a life stage that is not as vulnerable as the *in utero* period or infancy; (2) challenges of recruiting young children and obtaining biospecimens; and (3) difficulties of assessing diet in young children.

The second research gap is that the majority of studies to date have considered single biological pathways, which overlooks the contribution of multiple mechanisms and interactions among them [124]. Integrative multi-‘omics studies, despite being a relatively new concept with no established guidelines on best practices, hold great potential to untangle key mechanisms underlying DOHaD of childhood obesity and aligns with the goals of precision nutrition, discussed in the next section.

Finally, the third gap was that none of the studies were able to examine the entire spectrum of nutrition biological mechanisms underlying obesity-related outcomes in order to ascertain temporality and causality among each link across the pathway of interest.

### 5.2. Future Opportunities for Multi-‘Omics Networks in Precision Nutrition/Health

In 2015, President Barack Obama launched the Precision Medicine Initiative, emphasizing a revolutionary new research effort to improve health and treat disease by focusing on individual-specific treatment approaches. As mentioned in their Mission Statement, the “one-size-fits-all” approach can be very successful for some patients but not for others. Perhaps precision medicine will enable and empower health care providers to tailor treatment and prevention strategies to an individual’s unique characteristics [27]. Personalized or precision nutrition is expected to become the goal of future health care, and multi-‘omics networks is the approach that shows huge potential to address individual differences in diagnosis and treatment, not only in adults, but also in children [125]. This combined information provided by multi-‘omics networks should not only determine the genetic susceptibility of the individual, but also monitors this person’s real-time physiological states [125]. Driven by technological advances that have made cost-effective, high-throughput analysis of biologic molecules possible, integration of multiple types of ‘omics data can be used to elucidate potential causative changes that lead to disease state, or the treatment targets [126].

Beyond the capacity of multi-‘omics to provide a more holistic view of physiology, it also has potential to provide researchers with insight on modifiability of disease mechanisms, from the original cause of disease (genetic, metabolomics, environmental, or developmental) to the functional consequences or relevant interactions [127]. Scientists have grouped current multi-‘omics network studies into three categories, “genome first,” “phenotype first,” and “environment first,” depending on the initial focus of the investigation [127]. The genome first approach seeks to determine the mechanisms by which GWAS loci contribute to disease; the phenotype first approach seeks to understand the pathways contributing to disease without centering the investigation on a particular locus; and the environment first approach examines the way environment perturbs pathways or interacts with genetic variation [127]. The former focuses on non-modifiable genetic determinants of disease, whereas the latter represents potentially modifiable disease mechanisms that may be linked to specific behavioral risk factors. We propose that by integrating genetics, epigenomics, and metabolomics using unsupervised dimension reduction approaches such as multi-‘omics networks [128,129], it will be possible to identify biological pathways linking early life nutritional exposures to obesity-related health outcomes, while also enabling assessment of modifiability of the pathways based on composition of the networks. Specifically, if a network of interest is composed predominantly of genetic SNPs, then the mechanism is likely not modifiable. On the other hand, if a network is largely composed of metabolites, then this suggests that this network may be amenable to dietary interventions given the responsibility of metabolites to environmental factors.

The integrative frameworks we introduced in Section 1, in conjunction with the literature reviewed herein call attention to the necessity of well-thought longitudinal study design that evaluates diet, ‘omics, and obesity-related health outcomes in the same sample. Moreover, a one-size-fits-all approach may not be sufficient to identify risk factors and mechanisms given marked inter-individual variability in the etiology of multifactorial complex diseases [11]. We purport that developmental and life course frameworks offer ways to conceptualize integrative multi-‘omics approaches that shed light on both “behind the scenes” mechanisms, as well as potential novel biomarkers of childhood obesity and obesity-related disease. Ultimately, these endpoints will unveil potential avenues for effective and timely preventive efforts.

### 5.3. Summary and Final Conclusion

The increasing trends in childhood obesity over the last two decades are alarming, as youth with chronic conditions will enter adulthood with several years of disease duration, difficulty in treatment, and greater risk of early complication [130,131]. While population-level strategies to reduce obesity rates are valuable (e.g., reduce intake of unhealthy fats and refined carbohydrates, increase physical activity levels), childhood obesity rates are stagnant and growing evidence suggests that a one-size-fits-all approach may not be enough given marked inter-individual variability in physiology, ranging from DNA sequence to behavior pattern. Despite the need to establish consensus guidelines for data processing and analysis, ‘omics technologies—both individually and in combination with DOHaD knowledge and life course epidemiology framework—are promising tools to better understand disease etiology—an effort that is already underway in type 2 diabetes research [132,133,134,135]. Given that an ongoing challenge of DOHaD research revolves around how one might untangle the effects of a specific exposures on disease risk, ‘omics analyses may present a rich opportunity to identify shared and overlapping biological pathways and ultimately, to link these pathways to modifiable environmental exposures or behaviors as one potential route to the holy grail of primary prevention.

Knowledge alone, however, is not sufficient to change behaviors [136]. With multi-‘omics information available, we have not completely figured out how to efficiently incorporate dietary advice into behavior changes. Özdemir and Kolker [137] proposed a nested innovation management and ethics oversight system for precision nutrition application. Based on the model, they expounded the requirements of three layers of knowledge in order to practice precision nutrition successfully, from the inside out, “technical knowledge”, “social knowledge”, and “knowledge-on-knowledge” [137]. At the current stage, there are still many unknown on the first layer. With an emphasis on precision nutrition/health in the future, our ultimate goal will be to build upon multi-‘omics networks foundation, utilizing social knowledge and following appropriate ethical codes, to help individuals to achieve better health outcomes and improve their quality of lives.

## Figures and Tables

**Table 1 metabolites-10-00235-t001:** Summary of studies that demonstrated potential completed nutrition, ‘omics, and childhood obesity risk pathways.

Types of ‘Omics	Author/s	Literature Demonstrated Potential Completed Pathways
Nutrigenomics		
*In utero*	Vincet et al. [34], Gil-Campos et al., [35]	MED diet and LFLC diet → *Apo E*, *Apo B* → childhood obesity
	Cook & Fletcher [36]; Tanaka et al., [37]	Essential FA intake → *FADS2* → offspring birth weight
*Infancy*	Miralles et al., [38]; Ahima and Flier., [39]; Elias et al., [39]	Breastfeeding → *LEP*, *NPY* and *POMC* → infant fat storage
	Ordovas et al., [40]; Pérusse & Bouchard., [41]; Hainer et al. [42]	Dietary fat and cholesterol intake → Apo and LDL-receptor genes → infant body weight and body fat
*Early-childhood*	Qi et al., [43]	Dietary protein intake → *FTO* → BMI
Epigenetics		
*In utero*	Lane et al., [44]; Liu et al. [45]	Paternal obesity → epigenetics modifications on *SGCE/PEG10* and *MEG3* DMRs → offspring birth weight
*Infancy*	Crujeiras et al., [46]; Ruchat et al., [47]; Lesseur et al., [48]	Breastfeeding → blood *LEP* promoter DNA methylation levels → SGA and body fat
*Early-childhood*	N/A	N/A
Metabolomics		
*In utero*	Klebanoff & Keim [49]	Caffeine intake → paracanthine metabolite concentrations → offspring BMI
	Donahue et al., [50]; Rauschert et al., [51]	LC-PUFA intake → omega-3 fatty acids metabolites → offspring weight status
*Infancy*	Kirchberg et al., [52]; Weber et al., [53]	Formula feeding → BCAAs metabolites (such as valine, leucine, and isoleucine) → elevated infant insulin and IGF-1 levels
*Early-childhood*	Michaliszyn et al., [54]; Zhao et al., [55]	Types and amounts of protein intake → BCAAs metabolites → weight status

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
