# Peer review of "Precision Nutrition and Childhood Obesity: A Scoping Review"

_metabolites, 2020, doi:10.3390/metabo10060235_

Round 1
Reviewer 1 Report
Dear authors
Initially, I would like to congratulate the authors for addressing such an important and relevant topic.
The introduction is very interesting but long, I suggest reducing it.
About the methods, it is said that searches were made on google scholar and pubmed engines (pubmed is not a database, the database is medline). In this context, I suggest that terms mesh be used for searching the medline, and that articles be searched for in EMBASE too, minimally. I suggest that specific repository bases of endocrinology be added. Google scholar does not have indexing criteria, so if searches are maintained on this base, there must necessarily be an assessment of the quality of the articles found. I suggest mentioning which study designs were considered, minimally, given that population studies can be a very vague term.
Regarding the results and discussion, before starting the discussion directly, I suggest at least informing how many articles were found and how many were reviewed.
The discussion is very interesting, however, some important authors were not mentioned. As the research question is very comprehensive, the discussion naturally becomes superficial. I suggest evaluating whether the work should not be divided into the multiple points raised and each one detailed.
The table presented at the end, with acyclic diagrams is very interesting, congratulations.
Again congratulations on the work. It is a great narrative review for someone who doesn’t know the subject to introduce themselves to the topic.
Author Response
Thank you for the opportunity to revise and re-submit our manuscript. We appreciate your careful review and specific comments that we respond to, in turn, below.
Reviewer #1:
Comment 1: The introduction is very interesting but long, I suggest reducing it.
Response: Thank you for this comment. We have now shortened the Introduction to half the previous length as you can see in the highlighted section.
Comment 2: About the methods, it is said that searches were made on google scholar and pubmed engines (pubmed is not a database, the database is medline). In this context, I suggest that terms mesh be used for searching the medline, and that articles be searched for in EMBASE too, minimally. I suggest that specific repository bases of endocrinology be added. Google scholar does not have indexing criteria, so if searches are maintained on this base, there must necessarily be an assessment of the quality of the articles found. I suggest mentioning which study designs were considered, minimally, given that population studies can be a very vague term.
Response: Thank you for this valuable suggestion. We agree that more detailed information should be included in the Methods section, especially around the selection criteria of studies. For this review, due to the limited literature, we selected population-based studies more based on years they were published (within the last 10 years), instead of specific study designs (cross-sectional, case control and prospective longitudinal studies all are included). In terms the search databases used, we did not limit our Medline search in MeSH databases, hoping to have a research collection as comprehensive as possible. We did use EMBASE and DynaMed databases to expand our research arena. Thank you for reminding us to include these information. To address the reviewer’s concern, we have revised this to read as follows:
“We elected to conduct a scoping review because the application of precision nutrition in childhood obesity prevention and treatment is relatively recent, underscoring the importance of identifying knowledge gaps to expedite further studies in this field. Evidence from original research included in this scoping review was identified using Google Scholar, Medline, EMBASE and DynaMed databases. First, we conducted a comprehensive search effort to gather presenting materials that are relevant to the main objective of this review — the potential associations between nutrition, ‘omics, and childhood obesity. We then considered population-based, peer-reviewed studies that were conducted and published within the last 10 years, including observational studies with a range of designs (cross-sectional, case control and prospective longitudinal studies). We did not limit dates or types (in vivo/in vitro/human studies) for literature cited to interpret the associations relating ‘omics evidence of in utero, infancy, and early-childhood exposure to mid-childhood obesity risks. Key search terms included “DOHaD theory,” “DOHaD,” “thrifty theory,” “maternal/in utero nutrition exposure,” “BMI rebound,” “infancy, breastfeeding,” “infancy, nutrition intake,” “early-childhood,” “early-childhood, nutrition intake,” “early-childhood, dietary behavior,” “precision nutrition/medicine,” “omics networks,” “multi-‘omics networks,” “childhood overweight/obesity” in combination with any of “nutrigenomics,” “nutrition and genetic interactions,” “epigenetics,” “DNA methylation,” “methylome,” “histone modification,” “metabolites,” or “metabolome”. To comprehensively identify both current limitations and future research opportunities, we expanded the literature search to include papers considering relevant methods, including cohort selection, data collection, statistical analysis, underlying multi-‘omics networks. Our search identified 67 peer-reviewed articles (not include method related literature). Among those, 23 articles were reviewed and discussed in detail and contributed to our focus on linkages between nutrition exposure —> ‘omics —> childhood obesity pathways (Table 1). Summary and discussion of reviewed articles were organized and presented by types of ‘omics, e.g., nutrigenomics, epigenomics and metabolomics, Within each ‘omics theme, results were aggregated by sensitive periods (in utero, infancy, and early-childhood). ”
Comment 3: Regarding the results and discussion, before starting the discussion directly, I suggest at least informing how many articles were found and how many were reviewed.
Response: Thank you for this valuable suggestion. We have inserted the following sentence at the end of the Methodology section: “Our search identified 67 peer-reviewed articles. Among those, 23 articles were reviewed and discussed in detail and contributed to the proposed potential nutrition exposure —> ‘omics —> childhood obesity pathways (Table 1).”
Comment 4: The discussion is very interesting, however, some important authors were not mentioned. As the research question is very comprehensive, the discussion naturally becomes superficial. I suggest evaluating whether the work should not be divided into the multiple points raised and each one detailed.
Response: We are pleased that the reviewer finds our review to be comprehensive and interesting. Regarding the Discussion, we feel that we have done precisely what this reviewer is suggesting by discussing the literature with respect to specific ‘omics fields, and specific life stages within which each ‘omics field is used to study DOHaD mechanisms. Further, we describe findings from key studies within each section and point out limitations of each. Thus, unless the Editor feels strongly otherwise, we have retained our current format for the Discussion.
Reviewer 2 Report
The scoping review by Wu et al. provides a thorough overview of nutrigenetics/genomics, epigenetic/genomics, and metabolomic studies examining obesity-related outcomes. This review has an important focus, providing a well-structured overview of omics studies examining obesity in three sensitive periods of early life. The authors identify current research gaps and limitations and make knowledgeable suggestions for future studies. However, there are a few minor issues that should be addressed before the manuscript is accepted for publication:
Introduction
Line 49 – should read over ‘the’ past two decades
Line 130 – should read ‘the’ development
Nutrigenomic section
The authors need to highlight that this section discusses both nutrigenomic and nutrigenetic studies. Studies which examine how genetic variants change response to diet are nutrigenetic studies, nutrigenomics studies examine how nutrients affect gene expression (as described at the start of the nutrigenomics section). Nutrigenetics and nutrigenomics are not interchangeable terms, these studies examine nutrient-gene interactions from different directions, and it is important that the authors distinguish this. It would be helpful if nutrigenetics is defined alongside nutrigenomics at the start of this section, and which studies are nutrigenomics vs. nutrigenetics studies is highlighted throughout.
Line 202 – should read ‘genes that respond to diet and nutrition…’
Epigenetics section
Line 391 – Review by Reynolds et al – the description of this review is vague; can the authors give more details of what ‘certain’ genes?
Line 437 – Unclear sentence – maybe should be ‘is consequence of mode of feeding’ not as a consequence?
Line 460 – EWAS has already been defined
Line 472 - MLLT4 unitalicized?
Metabolomics section
Line 604 – unclear what a mid-childhood age is, would be more helpful if authors included the actual mean age or age range.
Author Response
Thank you for the opportunity to revise and re-submit our manuscript. We appreciate your careful review and specific comments that we respond to, in turn, below.
Reviewer #2:
INTRODUCTION
Comment 1: Line 49 – should read over ‘the’ past two decades
Response: Thank you for catching this. However, in response to Reviewer #1’s comment about the Introduction being long, we have removed this sentence from the Introduction.
Comment 2: Line 130 – should read ‘the’ development
Response: Thank you. We have made the correction as suggested.
“…… Secondly, while many factors are involved in the development of excess adiposity, we describe specific mechanisms identified in the literature as potential pathways linking nutrition to overweight/obesity during childhood.
NUTRIGENOMIC Section
Comment 3: The authors need to highlight that this section discusses both nutrigenomic and nutrigenetic studies. Studies which examine how genetic variants change response to diet are nutrigenetic studies, nutrigenomics studies examine how nutrients affect gene expression (as described at the start of the nutrigenomics section). Nutrigenetics and nutrigenomics are not interchangeable terms, these studies examine nutrient-gene interactions from different directions, and it is important that the authors distinguish this. It would be helpful if nutrigenetics is defined alongside nutrigenomics at the start of this section, and which studies are nutrigenomics vs. nutrigenetics studies is highlighted throughout.
Response: Thank you for this valuable suggestion. We now emphasize this point in the opening paragraph on nutrigenomics:
“Nutrigenomics is the holistic study of gene-diet interactions. This field followed that of the older concept of nutrigenetics, which has been sensationalized in popular press as “feeding your DNA,” or “eating for your genes.” Nutrigenetics scrutinizes the effect of polymorphisms of single genes and how these differences affect one’s metabolism and thus, carries implications for the most appropriate foods to consume (or avoid). On the other hand, nutrigenomics is concerned with the potential for diet to change gene function and expression via alterations of the epigenome. Indeed, the goal of nutrigenomics is to unravel the interaction between genetics and dietary intake by bringing together bioinformatics, nutrition, molecular biology, genetics, genomics, epidemiology, and molecular medicine [26]. In this review, we focus on nutrigenomics and note that because most studies to date have only assessed diet and the epigenome at a single point in time, it has not been possible to untangle whether changes in diet cause changes in the epigenome. Thus, current literature simply provides evidence for an association between dietary intake and genome-wide differences in DNA methylation.”
Comment 4: Line 202 – should read ‘genes that respond to diet and nutrition…’
Response: Thank you. We have made the correction as suggested.
“……1) identify the genes that respond to diet and nutrition modifications; ….…
EPIGENETICS Section
Comment 5: Line 391 – Review by Reynolds et al – the description of this review is vague; can the authors give more details of what ‘certain’ genes?
Response: Thank you for this valuable suggestion. We have inserted the following sentence in the paragraph.
“As we mentioned in the Nutrigenomics section, one of the major limitations of ‘omics research is that very few human studies have employed the appropriate study design or sufficient length of follow-up to examine the dietary exposure, ‘omics, and health outcome in the same population. In a review conducted by Reynolds et al. [80], the authors thoroughly summarized current evidence on the association of maternal exposures with DNA methylation of certain genes, and the association of DNA methylation with health outcomes. Genes of interested included but were not limited to: IGF2, ADIPOQ, NR3C1, SLC6A4, MEG3, CYP1A1. However, these genes were not then linked to modifiable exposures, like diet, or to important offspring health outcomes.”
Comment 6: Line 437 – Unclear sentence – maybe should be ‘is consequence of mode of feeding’ not as a consequence?
Response: Thank you. We have made the correction as suggested.
“Among infants, a key area of interest regarding epigenetics is the consequence of mode of feeding (i.e., breastfeeding duration and exclusivity, formula vs. breast milk, and breast- vs. or bottle-feeding), with respect to birth size. Another area of influence highlights the relationships between modifications of epigenetic and cardiometabolic health outcomes.”
Comment 7: Line 460 – EWAS has already been defined
Response: Thank you for paying attention to the detail. We have made the correction as recommended.
“……Rzehak et al. conducted a EWAS based on a subset of 374 children out of 543 children aged 5.5 years from European Childhood Obesity Trial Study (CHOP) cohort …”
Comment 8: Line 472 – MLLT4 unitalicized?
Response: Thank you for the kind reminder. We have corrected this mistake.
“……MLLT4 (myeloid/lymphoid or mixed-lineage leukaemia) and CILP (Cartilage intermediate layer protein……”
METABOLOMICS Section
Comment 9: Line 604 – unclear what a mid-childhood age is, would be more helpful if authors included the actual mean age or age range.
Response: Thank you for this valuable suggestion. We have included those information as follows:
“……Following off of these findings, Perng et al. explored metabolomics profiles of childhood obesity and metabolic risk in diverse populations, including in a cohort of Mexican adolescents (with a median age lying in the mid-childhood range: 7.7 years (range: 6.7 to 10.6)), wherein the authors identified metabolites involved in lipid, amino acid, and carbohydrate metabolism as correlates of a metabolic syndrome risk score [119];……”
Reviewer 3 Report
The authors present a scoping review of existing studies that intended to unravel biological pathways that underlie the link between nutritional exposures and obesity during the early years of life. To have a comprehensive picture of the research done so far in this specific field is particularly relevant, not only because there is a serious growing concern about the increased prevalence of childhood obesity, but also because the first years of life are especially sensitive to environmental exposures. Thus, understanding the biological mechanisms through omics that may influence the course of childhood obesity can be of great help in the development of prevention and risk assessment tools.
In terms of the choice of omics presented there is surprisingly no mention of the gut microbiome. This may be one of the most recently studied omics in the research field of obesity and nutrition with a clear role in intra- and inter-individual variability. If the authors chose to not focus on this omics they should explain why and mention it across the review, in particular page 5. In addition, page 8 lines 298 – 303 “surprising that fad [should be fat] diets often engender very different responses in different individuals”. Beyond their own genes (host), microbial gene pool also drives a large part of these different responses.
The authors should describe, if they exist multi-omics studies on this topic.
It is not always clear when the studies are included/excluded. It seems that the authors selected the ones focusing on omics, nutritional exposures and obesity during the early years of life, however many studies presented do not include nutrition.
Finally, the review should be shortened in general, for example by removing text going off topic such as the whole page on epigenetic technology development (page10 lines 348 -380).
Specific comments:
Just as in the introduction, where the authors explain that the review focuses on research made on three sensitive periods (in utero, infancy and early-childhood), the methodology could contain a brief paragraph explaining that the resulting studies will be classified and presented by 1/ type of omics layer and 2/ sensitive period addressed. It would ease the reader’s transition to the results.
Method: The review could benefit from some key search terms being more disaggregated. In particular, instead of looking for “DOHaD theory” as a whole, “DOHaD” AND “DOHaD theory” would provide a more extensive search result. The same could apply to “multi-‘omics networks”, where the authors could consider “omics networks” or “multi-omics” as well. Also, the authors could add “methylome” and “metabolome” to the omics search terms in order to complement “DNA methylation” and “metabolites”.
Some verb tenses should be checked in order to be consistent with the tens used along the review (normally past tense).
The following sentences result confusing or difficult to follow:
Line 260 à “… and its circulating levels signals nutritional status and energy storage levels to feeding centers in the hypothalamus and other central areas”.
Line 437 à “Among infants, a key area of interest regarding epigenetics as a consequence of mode of feeding (i.e., breastfeeding duration and exclusivity, formula vs. breast milk, and breast- vs. or bottle-feeding) [88], with respect to birth size.”.
Is the sentence lacking the verb? It is difficult to follow what the authors try to say.
Line 448 to 454 à “… and they observed that firstly, there was a strong correlation between…; and secondly, higher LEP promoter DNA methylations observed in small-for-gestational-age (SGA) infants only, …”.
This sentence feels grammatically incoherent. It should be something like “they observed that firstly, there was…; and secondly, there was higher LEP promoter DNA methylation in small-for-gestational-age (SGA) infants only, …”.
Author Response
Thank you for the opportunity to revise and re-submit our manuscript. We appreciate your careful review and specific comments that we respond to, in turn, below.
Reviewer #3:
Comment 1: In terms of the choice of omics presented there is surprisingly no mention of the gut microbiome. This may be one of the most recently studied omics in the research field of obesity and nutrition with a clear role in intra- and inter-individual variability. If the authors chose to not focus on this omics they should explain why and mention it across the review, in particular page 5.
Response: Thank you for your comment and we concur with the reviewer that this is an important issue. Indeed, in writing the review, we carefully considered this exact notion and, in the end, decided not to include the gut microbiome f for two reasons. First, the gut microbiome is not conventionally considered as part of the ‘omics cascade [c.f., Buescher & Driggers, 2016; Beale, Karpe & Ahmed, 2016]. Secondly, the microbiome may actually be part of our outcome of interest in this review given that weight status has potential to modify the gut microbiome given the endocrine properties of adipose tissue and the effects of adipose-derived hormones (e.g., inflammatory cytokines, adipokines like leptin, ghrelin, and adiponectin) – c.f., Dandoval Endocrinology 2014; 155(3):653-655 on the role of leptin in regulating microbiota composition) on physiological pathways that can influence microbiome composition).
Some readers may have the same question while reading this manuscript, so we have inserted the following at the end of Section 3. ‘Omics Fields of Interest in this Review:
“Of note, despite increasing evidence of a role for the gut microbiome as a modifier or mediator to ‘omics mechanisms, we did not include this topic in the present review since the microbiome is not typically considered part of the ‘omics cascade, and because composition of the gut microbiota can be affected by weight status and thus, may be part of the outcome rather than a mechanism.”
Comment 2: In addition, page 8 lines 298 – 303 “surprising that fad [should be fat] diets often engender very different responses in different individuals”. Beyond their own genes (host), microbial gene pool also drives a large part of these different responses.
Response: Thank you for your attention to this detail, but we did not misspell fad diets (diets that are popular for a time, but may not adhere to an evidence-based dietary recommendations, e.g. Atkins Diet, South Beach Diet, Paleo Diet). The hypothetical explanation we are trying to make is that based on previous literature, so-called fad diets may work for some group of people while have no health benefits for others, due to inter-individual genetic differences, To address the reviewer’s concern, we have reworded this sentence as follows:
“Thus, it is not surprising that short-lived, popular diets often engender very different responses in different individuals”
Please refer to our response above, to Comment 1, for the rationale of not including the microbial gene pool as a factor underlying differences in responses to different diets in this review.
Comment 3: The authors should describe, if they exist multi-omics studies on this topic.
Response: Thank you for this comment. To our knowledge, there are no existing multi-‘omics studies that focus on childhood obesity as the health outcome, which is one of our motivations to conduct this scoping review. There is multi-‘omics studies of type 2 diabetes among adults.
To avoid confusion, we have emphasized this research gap in the Introduction: 1.2. Section.
“While FOAD focused on the pre-, peri-, and immediate postnatal periods (typically spanning birth through the first two years of life) as key life stages of interest for “programming” future health, DOHaD recognized the relevance of critical/sensitive periods beyond infancy including: early childhood around the time of the adiposity rebound [17]; late childhood/early adolescence when the number of adipocytes an individual possesses is thought to be established for life [18]; and the pubertal transition [19], a time of rapid growth and hormonal fluctuation. Pregnancy is now hypothesized as another sensitive period not only for the fetus but also for the gravida given findings that adverse pregnancy outcomes (e.g., gestational diabetes, hypertensive disorders, preterm birth) presage a woman’s as well as her offspring’s risk for cardiometabolic disease down the road [20, 21]. However, to our knowledge, there is no existing population-based epidemiological multi-‘omics studies that focus on childhood obesity as the health outcome, which could expand our knowledge on this field.”
Comment 4: It is not always clear when the studies are included/excluded. It seems that the authors selected the ones focusing on omics, nutritional exposures and obesity during the early years of life, however many studies presented do not include nutrition.
Response: Thank you for this valuable suggestion. We agree that more inclusion/exclusion criteria related information should be included in the Methods section. The updated paragraph is as follows:
“We elected to conduct a scoping review because the application of precision nutrition in childhood obesity prevention and treatment is relatively recent, underscoring the importance of identifying knowledge gaps to expedite further studies in this field. Evidence from original research included in this scoping review was identified using Google Scholar, Medline, EMBASE and DynaMed databases. First, we conducted a comprehensive search effort to gather presenting materials that are relevant to the main objective of this review — the potential associations between nutrition, ‘omics, and childhood obesity. We then considered population-based, peer-reviewed studies that were conducted and published within the last 10 years, including observational studies with a range of designs (cross-sectional, case control and prospective longitudinal studies). We did not limit dates or types (in vivo/in vitro/human studies) for literature cited to interpret the associations relating ‘omics evidence of in utero, infancy, and early-childhood exposure to mid-childhood obesity risks. Key search terms included “DOHaD theory,” “DOHaD,” “thrifty theory,” “maternal/in utero nutrition exposure,” “BMI rebound,” “infancy, breastfeeding,” “infancy, nutrition intake,” “early-childhood,” “early-childhood, nutrition intake,” “early-childhood, dietary behavior,” “precision nutrition/medicine,” “omics networks,” “multi-‘omics networks,” “childhood overweight/obesity” in combination with any of “nutrigenomics,” “nutrition and genetic interactions,” “epigenetics,” “DNA methylation,” “methylome,” “histone modification,” “metabolites,” or “metabolome”. To comprehensively identify both current limitations and future research opportunities, we expanded the literature search to include papers considering relevant methods, including cohort selection, data collection, statistical analysis, underlying multi-‘omics networks. Our search identified 67 peer-reviewed articles (not include method related literature). Among those, 23 articles were reviewed and discussed in detail and contributed to our focus on linkages between nutrition exposure —> ‘omics —> childhood obesity pathways (Table 1). Summary and discussion of reviewed articles were organized and presented by types of ‘omics, e.g., nutrigenomics, epigenomics and metabolomics, Within each ‘omics theme, results were aggregated by sensitive periods (in utero, infancy, and early-childhood). ”
Many studies presented did not include nutrition, given the likely challenges of addressing nutrition exposure, ‘omics, and health outcomes in a single study. Therefore, we aimed to connect the dots among these three factors. Potential pathways included in Table 1 are based on two types of population-based studies (they study could be longitudinal, or cross-sectional, or case-control): 1). articles presenting the association between nutrition exposures and ‘omics; and 2) articles presenting the association between ‘omics and childhood obesity. We have already discussed this limitation in section 5.1. Summary of research gaps from the original version of the manuscript, as follows:
“Finally, the third gap was that none of the studies were able to examine the entire spectrum of nutrition biological mechanisms underlying obesity-related outcomes in order to ascertain temporality and causality among each link across the pathway of interest.”
Comment 5: The review should be shortened in general, for example by removing text going off topic such as the whole page on epigenetic technology development (page10 lines 348 -380).
Response: In response, we have shortened this paragraph to approximately half of the original one since the development of epigenetic technology has been discussed widely elsewhere. We have kept essential information and citations in case readers want to expand their knowledge on each specific method. The new paragraph is as follows:
“Since these initial studies that employed candidate gene approaches, technology to investigate epigenetic changes has markedly improved afterwards. Currently, the three major approaches to assessment of DNA methylation used in epidemiological research are global DNA methylation, gene-specific DNA methylation, and epigenome-wide studies of DNA methylation at the resolution of single CpG sites. In studies of pediatric populations, DNA methylation has largely been assayed from circulating leukocytes given challenges of more invasive sampling procedures to obtain target tissues (e.g., adipose tissue). In earlier studies published through the early 2010s, a large emphasis was placed on associations of early nutritional exposures and obesity-related outcomes in relation to global DNA methylation – an indicator of total methyl group content within an individual’s genome, and a metric of genomic stability. However, a limitation to both global DNA methylation and candidate gene studies is that the etiology of complex diseases (such as obesity and related conditions) likely involves several biological pathways. This notion gave rise to use of epigenome-wide association studies (EWAS) to identify multiple differentially-methylated regions of the genome to gain a more holistic understanding of epigenetic alterations associated with various exposures or health outcomes. In the subsequent sections, we discuss existing literature as they pertain to each of the three methods of assessing DNA methylation (global DNA methylation, candidate gene DNA methylation, and EWAS).”
Comment 6: Just as in the introduction, where the authors explain that the review focuses on research made on three sensitive periods (in utero, infancy and early-childhood), the methodology could contain a brief paragraph explaining that the resulting studies will be classified and presented by 1/ type of omics layer and 2/ sensitive period addressed. It would ease the reader’s transition to the results.
Response: Thank you for this valuable comment. We have inserted this statement at the end of the Method section. Please see the revised text below:
“We elected to conduct a scoping review because the application of precision nutrition in childhood obesity prevention and treatment is relatively recent, underscoring the importance of identifying knowledge gaps to expedite further studies in this field. Evidence from original research included in this scoping review was identified using Google Scholar, Medline, EMBASE and DynaMed databases. First, we conducted a comprehensive search effort to gather presenting materials that are relevant to the main objective of this review — the potential associations between nutrition, ‘omics, and childhood obesity. We then considered population-based, peer-reviewed studies that were conducted and published within the last 10 years, including observational studies with a range of designs (cross-sectional, case control and prospective longitudinal studies). We did not limit dates or types (in vivo/in vitro/human studies) for literature cited to interpret the associations relating ‘omics evidence of in utero, infancy, and early-childhood exposure to mid-childhood obesity risks. Key search terms included “DOHaD theory,” “DOHaD,” “thrifty theory,” “maternal/in utero nutrition exposure,” “BMI rebound,” “infancy, breastfeeding,” “infancy, nutrition intake,” “early-childhood,” “early-childhood, nutrition intake,” “early-childhood, dietary behavior,” “precision nutrition/medicine,” “omics networks,” “multi-‘omics networks,” “childhood overweight/obesity” in combination with any of “nutrigenomics,” “nutrition and genetic interactions,” “epigenetics,” “DNA methylation,” “methylome,” “histone modification,” “metabolites,” or “metabolome”. To comprehensively identify both current limitations and future research opportunities, we expanded the literature search to include papers considering relevant methods, including cohort selection, data collection, statistical analysis, underlying multi-‘omics networks. Our search identified 67 peer-reviewed articles (not include method related literature). Among those, 23 articles were reviewed and discussed in detail and contributed to our focus on linkages between nutrition exposure —> ‘omics —> childhood obesity pathways (Table 1). Summary and discussion of reviewed articles were organized and presented by types of ‘omics, e.g., nutrigenomics, epigenomics and metabolomics, Within each ‘omics theme, results were aggregated by sensitive periods (in utero, infancy, and early-childhood). ”
Comment 7: Method: The review could benefit from some key search terms being more disaggregated. In particular, instead of looking for “DOHaD theory” as a whole, “DOHaD” AND “DOHaD theory” would provide a more extensive search result. The same could apply to “multi-‘omics networks”, where the authors could consider “omics networks” or “multi-omics” as well. Also, the authors could add “methylome” and “metabolome” to the omics search terms in order to complement “DNA methylation” and “metabolites”.
Response: Thank you for this valuable suggestion. We agree that more information should be included in the Methods section. The updated paragraph can be found in our response to Comment 6.
Comment 8: Some verb tenses should be checked in order to be consistent with the tens used along the review (normally past tense).
Response: Thank you. We have double checked the verb tenses throughout the manuscript, and made corrections accordingly.
Comment 9: Line 260 “… and its circulating levels signals nutritional status and energy storage levels to feeding centers in the hypothalamus and other central areas”.
Response: Thank you. We have modified the sentence as follows:
“ ……Leptin is a hormone primarily produced and secretly by the adipose tissue, and its circulating levels signal nutritional status and energy storage levels to feeding centers in the hypothalamus and other central areas.”
Comment 10: Line 437 “Among infants, a key area of interest regarding epigenetics as a consequence of mode of feeding (i.e., breastfeeding duration and exclusivity, formula vs. breast milk, and breast- vs. or bottle-feeding) [88], with respect to birth size.” Is the sentence lacking the verb? It is difficult to follow what the authors try to say.
Response: Thank you for attending to the details. We have made the correction as suggested.
“Among infants, a key area of interest regarding epigenetics is the consequence of mode of feeding (i.e., breastfeeding duration and exclusivity, formula vs. breast milk, and breast- vs. or bottle-feeding), with respect to birth size. Another area of influence highlights the relationships between modifications of epigenetic and cardiometabolic health outcomes.”
Comment 11: Line 448 to 454 “… and they observed that firstly, there was a strong correlation between…; and secondly, higher LEP promoter DNA methylations observed in small-for-gestational-age (SGA) infants only, …”. This sentence feels grammatically incoherent. It should be something like “they observed that firstly, there was…; and secondly, there was higher LEP promoter DNA methylation in small-for-gestational-age (SGA) infants only …”.
Response: Thank you for pointing it out. We have modify the sentence as follows:
“…… For instance, Lesseur et al. collected blood samples at different time points from 39 mother-infant pairs. They observed a strong correlation between maternal blood LEP promoter DNA methylation and infant blood LEP methylation; and secondly, noted higher LEP promoter DNA methylations among small-for-gestational-age (SGA) infants only. Higher LEP promotor DNA methylation could result in lower circulating leptin levels, then affecting infant growth and development…….”

Round 2
Reviewer 2 Report
Thank you for addressing the comments.
Reviewer 3 Report
The authors generally put a big effort into addressing all our suggestions and they have made everything much clearer now. A small note in comment 9, the correction of a sentence, a grammar mistake was missed.
“ ……Leptin is a hormone primarily produced and secreted by the adipose tissue, and its circulating levels signal nutritional status and energy storage levels to feeding centers in the hypothalamus and other central areas.”
Secreted, instead of “secretly”